# Patient hesitancy in perioperative clinical trial enrollment during the COVID-19 pandemic

**Josue D. Chirinos**[1], **Isabella S. Turco**[2], **Raffaele Di Fenza**[1,2], **Stefano Gianni**[1,2], **Grant M. Larson**[2,3], **Joseph F. Swingle**[4], **Oluwaseun Akeju**[1,2☯], **Lorenzo Berra**[1,2,5☯]*

**1** Harvard Medical School, Boston, Massachusetts, United States of America, **2** Department of Anesthesia, Critical Care, and Pain Medicine, Massachusetts General Hospital, Boston, Massachusetts, United States of America, **3** Geisel School of Medicine at Dartmouth, Hanover, New Hampshire, United States of America, **4** Department of Sociology, Wellesley College, Wellesley, Massachusetts, United States of America, **5** Respiratory Care Services, Massachusetts General Hospital, Boston, Massachusetts, United States of America

☯ These authors contributed equally to this work.
\* lberra@mgh.harvard.edu

**Data Availability Statement:** All relevant data are within the paper and its Supporting Information files.

## Abstract

The COVID-19 pandemic has caused tremendous disruptions to non-COVID-19 clinical research. However, there has been little investigation on how patients themselves have responded to clinical trial recruitment during the COVID-19 pandemic. To investigate the effect of the COVID-19 pandemic on rates of patient consent to enrollment into non-COVID-19 clinical trials, we carried out a cross-sectional study using data from the Nitric Oxide/Acute Kidney Injury (NO/AKI) and Minimizing ICU Neurological Dysfunction with Dexmedetomi-dine-Induced Sleep (MINDDS) trials. All patients eligible for the NO/AKI or MINDDS trials who came to the hospital for cardiac surgery and were approached to gain consent to enrollment were included in the current study. We defined "Before COVID-19" as the time between the start of the relevant clinical trial and the date when efforts toward that clinical trial were deescalated by the hospital due to COVID-19. We defined "During COVID-19" as the time between trial de-escalation and trial completion. 5,015 patients were screened for eligibility. 3,851 were excluded, and 1,434 were approached to gain consent to enrollment. The rate of consent to enrollment was 64% in the "Before COVID-19" group and 45% in the "During COVID-19" group (n = 1,334, $P<0.001$) (RR = 0.70, 95% CI 0.62 to 0.80, $P<0.001$). Thus, we found that rates of consent to enrollment into the NO/AKI and MINDDS trials dropped significantly with the onset of the COVID-19 pandemic. Patient demographic and socioeconomic status data collected from electronic medical records and patient survey data did not shed light on possible explanations for this observed drop, indicating that there were likely other factors at play that were not directly measured in the current study. Increased patient hesitancy to enroll in clinical trials can have detrimental effects on clinical science, patient health, and patient healthcare experience, so understanding and addressing this issue during the COVID-19 pandemic is crucial.

**Funding:** LB receives salary support from K23 HL128882/NHLBI NIH as principal investigator for his work on hemolysis and nitric oxide. LB receives technologies and devices from Inhaled Nitric Oxide (NO) Therapeutics LLC, Masimo Corp. This study was supported through LB by the Reginald Jenney Endowment Chair at Harvard Medical School, by Sundry Funds at Massachusetts General Hospital, and by laboratory funds of the Anesthesia Center for Critical Care Research of the Department of Anesthesia, Critical Care, and Pain Medicine at Massachusetts General Hospital. The funders had no role in study design, data collection and analysis, decision to publish, or preparation of the manuscript.

**Competing interests:** I have read the journal's policy and the authors of this manuscript have the following competing interests: LB has filed a patent application on June 7, 2021 for NO delivery in COVID-19 disease: PCT application number: PCT/US2021/036269. This does not alter our adherence to PLOS ONE policies on sharing data and materials.

## Introduction

Since the World Health Organization (WHO) announced the rise of a novel coronavirus-related pneumonia in Wuhan, China on January 9, 2020 [1], the coronavirus disease 2019 (COVID-19) pandemic has dramatically altered the lives of virtually every human on the planet. According to the WHO, as of November 2022, there have been over 639 million confirmed cases of COVID-19 worldwide and over 6.6 million associated deaths [2]. Additionally, the COVID-19 pandemic has imposed substantial economic burdens, with early estimates in the tens of trillions of dollars in the United States alone [3].

The COVID-19 pandemic has also caused tremendous disruptions to non-COVID-19 clinical research. An April 2020 study showed that of over 1,000 clinical trial site personnel surveyed, 69% reported that the COVID-19 pandemic had affected their ability to conduct ongoing clinical trials and 78% reported that it had affected their ability to initiate new clinical trials [4–6]. Additionally, studies have reported decreases in clinical trial patient recruitment [4]. In this setting, there was a concomitant global decline in patient enrollment during the COVID-19 pandemic, with a 59% drop from the pre-COVID-19 baseline by the end of April 2020 [7].

Clinical trial disruptions and reduced patient enrollments have detrimental effects on the advancement of clinical science. Additionally, patients enrolled in clinical trials often receive new cutting-edge treatments, have more frequent health check-ups and medical care, and gain access to information about disease support groups and resources [8]. Thus, clinical trial disruptions and reduced patient enrollments into clinical trials can also negatively impact patient health, making these phenomena important topics of investigation.

Although there is a thorough understanding of the negative impact of the COVID-19 pandemic on non-COVID-19 clinical trials and patient recruitment, there has been little investigation on how patients themselves have responded to clinical trial recruitment during the COVID-19 pandemic. Given this gap in knowledge, the following study sought to investigate the effect of the COVID-19 pandemic on rates of eligible patient consent to enrollment into two non-COVID-19 clinical trials carried out at a large teaching hospital in Boston, Massachusetts. Using patient demographic and survey data, we also sought to investigate possible explanations for any observed changes in rates of consent to enrollment.

## Methods

### Patient screening and enrollment procedure for the NO/AKI and MINDDS trials

To investigate the effect of the COVID-19 pandemic on rates of eligible patient consent to enrollment into non-COVID-19 clinical trials, we performed a cross-sectional study. We used data from the Nitric Oxide/Acute Kidney Injury (NO/AKI) trial (NCT02836899) and the Minimizing ICU Neurological Dysfunction with Dexmedetomidine-Induced Sleep (MINDDS) trial (NCT02856594). The NO/AKI trial was carried out at a large teaching hospital in Boston, Massachusetts from May 2017 to June 2021, and the MINDDS trial was carried out at the same teaching hospital from March 2017 to August 2021. All patients who came to the hospital for cardiac surgery during the time frames of the clinical trials were screened for eligibility. Patients were excluded if they were found to be ineligible for the studies based on inclusion and exclusion criteria, which have been reported previously [9, 10], or if they were found to be enrolled in another clinical trial. For both trials, all remaining eligible patients were approached to gain consent to enrollment, and these patients either consented or refused.

Thus, the term hesitancy as used in this manuscript in the context of clinical trial enrollment was defined by the rate of patient refusal to enroll.

For the purposes of the current study, we assigned patients to either the "Before COVID-19" group or "During COVID-19" group based on the date that they were screened for entry into the NO/AKI or MINDDS trial. We defined "Before COVID-19" as the time between the start of the relevant clinical trial and the date when efforts toward that clinical trial were deescalated by the hospital due to COVID-19. Thus, "Before COVID-19" was defined as May 2017 to January 2020 for the NO/AKI trial and as March 2017 to March 2020 for the MINDDS trial. Similarly, we defined "During COVID-19" as the time between trial de-escalation and trial completion. Thus, "During COVID-19" was defined as February 2020 to June 2021 for the NO/AKI trial and as April 2020 to August 2021 for the MINDDS trial. Efforts toward the NO/AKI trial were deescalated before the MINDDS trial because there was an increased clinical demand for NO/AKI trial research staff during the pandemic. Additionally, the NO/AKI trial required use of devices prioritized for patients hospitalized for COVID-19, which limited the ability of the trial to proceed.

## Collection of patient enrollment, demographic, and survey data

To determine whether the COVID-19 pandemic had any effect on rates of eligible patient consent to enrollment into the NO/AKI and MINDDS trials, we collected enrollment data from clinical trial records. Data points included total number of consents and refusals to enrollment and total number of eligible patients approached for a given time period.

To characterize the populations of interest and investigate possible explanations for observed changes in rates of consent to enrollment into the NO/AKI and MINDDS trials, we collected demographic data from patient electronic medical records (EMRs). Data points included age, sex, race and ethnicity, and residential Zone Improvement Plan (ZIP) Code. We used ZIP Codes to determine three main indicators of socioeconomic status (SES) for patient residential areas. To do this, we first converted ZIP codes to ZIP Code Tabulation Areas (ZCTAs) using the ZIP Code to ZCTA Crosswalk tool [11]. Then, we used patient ZCTAs to determine median annual household incomes (MAHIs), median educational attainment levels (MEALs), and unemployment rates of patient residential areas [12].

Finally, to investigate another possible explanation for observed changes in rates of consent to enrollment, we collected survey data either in person or over the phone from patients enrolled in the NO/AKI trial. The survey consisted of 14 questions divided into three sections (informed consent, healthcare experience, and trial satisfaction), the latter two of which were meant to determine the impact of the NO/AKI trial on patient healthcare experience and assess patient satisfaction with their participation in the study. S1 Table provides a summary of all 14 questions asked in the patient surveys by section as well as the type of response required for each.

## Statistical analysis

Regarding continuous variables, we expressed patient ages in years and unemployment rates as percentages. Regarding categorical variables, we expressed patient rates of consent to enrollment and NO/AKI trial survey response frequencies as percentages. We expressed sex using male or female, race and ethnicity using four categories (non-Hispanic white, non-Hispanic Black, Hispanic, and Asian), MAHI using six categories (<$25,000, $25,000 to <$50,000, $50,000 to <$75,000, $75,000 to <$100,000, $100,000 to <$200,000, and ≥$200,000), and MEAL using six categories (less than high school, high school or equivalent, some college,

two-year degree, four-year degree, and graduate school). Finally, we summarized NO/AKI trial free response data qualitatively.

We compared differences in mean age and mean unemployment rates between the "Before COVID-19" and "During COVID-19" groups using two-sided t-tests. We compared enrollment inquiry responses, differences in sex frequencies, and differences in response frequencies to questions eight and nine of the NO/AKI trial survey between the two time frames using chi-square testing. For enrollment inquiry responses, we also calculated risk ratios, confidence intervals, and p-values to quantify the association between time frame and response frequency. We compared differences in race and ethnicity frequencies and differences in response frequencies to questions 12 and 13 of the NO/AKI trial survey between the two time frames using Fisher's exact test. For MAHI, MEAL, and questions 10 and 11 of the NO/AKI trial survey, we compared distributions of data from the two time frames to one another using the Levene test for homogeneity of variance to establish similarity. We then compared the MAHIs, MEALs, and response frequencies to questions 10 and 11 of the NO/AKI trial survey between the two time frames using the Mann Whitney U test. We considered p-values <0.05 to be statistically significant.

All statistical analyses were carried out using R [13], risk ratio statistics were calculated using the R packages epiR [14] and survival [15], Levene testing was carried using the R packages car [16] and carData [17], and all figures were produced using the R package ggplot2 [18].

### Ethics statement

This work was approved by the Mass General Brigham Human Research Committee (MGBHRC), the Institutional Review Board (IRB) of Mass General Brigham. All informed consent was obtained in written format.

## Results

### Patient screening data

Over the time periods studied, the NO/AKI and MINDDS trials screened a total of 5,015 patients that came to the hospital where the clinical trial was taking place for cardiac surgery. Of these patients, 3,851 were excluded due to ineligibility, enrollment in another study, or for other reasons. The remaining 1,434 patients were approached to gain consent to enrollment, and of these approached eligible patients, 793 consented to enrollment and 641 refused. A summary of the screening data is provided in Fig 1.

### Patient demographic data

Between the NO/AKI and MINDDS trials, the mean age was 68.81 years. Most patients were male, and the most frequent race and ethnicity was non-Hispanic white. Regarding residential SES indicators, the mean ZCTA MAHI was $95,392, the median ZCTA MEAL was some college, and the mean ZCTA unemployment rate was 4.11%. Sample sizes varied for each of the demographic characteristics studied due to missing data from patient EMRs. Patients with missing data were excluded from calculations. A summary of the demographic characteristics and residential SES indicators of all eligible patients approached to gain consent to enrollment into the NO/AKI and MINDDS trials is provided in Table 1.

### Patient enrollment data

Taking the enrollment data from the NO/AKI and MINDDS trials in aggregate, the rate of consent to enrollment among approached eligible patients dropped significantly with the

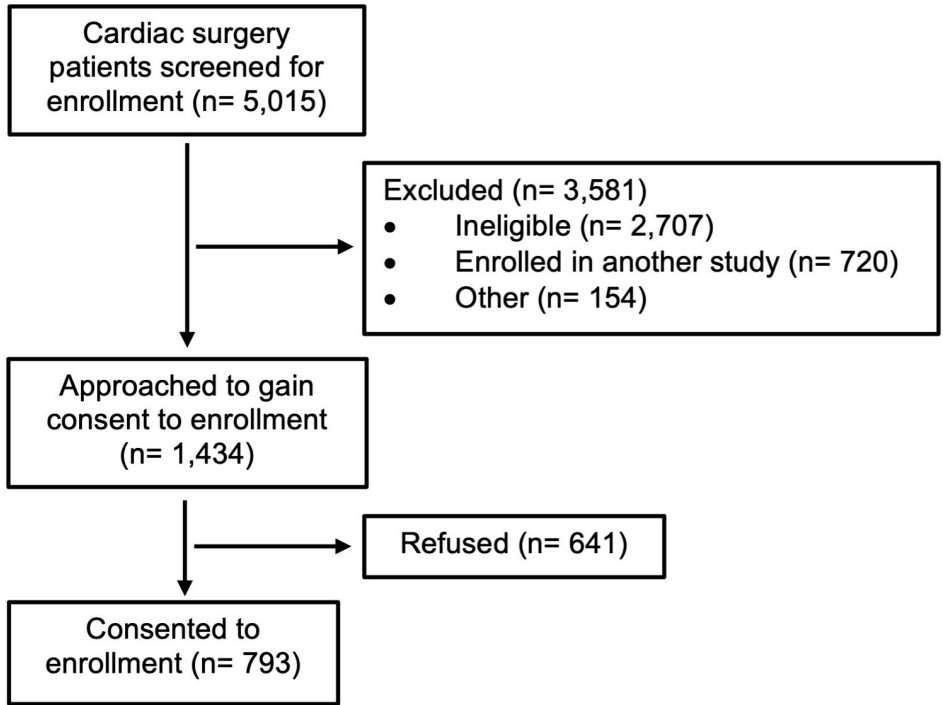

**Fig 1. Patient screening and enrollment for the NO/AKI and MINDDS trials.** All patients presenting for cardiac surgery were screened for enrollment. Patients were excluded from the trials if they were found to be ineligible based on inclusion or exclusion criteria, enrolled in another study, or for various other reasons.

onset of the COVID-19 pandemic. In the "Before COVID-19" group, the rate of consent to enrollment was 64% (n = 1,001), while in the "During COVID-19" group, it was 45% (n = 333), giving a difference in proportions of 19% (95% CI 12.88% to 25.12%). Importantly, this drop was statistically significant (n = 1,334, $P<0.001$). The risk ratio was 0.70 (95% CI 0.62 to 0.80, $P<0.001$), indicating that patients in the "During COVID-19" group were 30 percent less likely to consent to enrollment compared to patients in the "Before COVID-19" group. A summary of the aggregate patient enrollment data for the NO/AKI and MINDDS trials is provided in Fig 2.

### Analysis of patient demographic data

Taking the patient demographic data from the NO/AKI and MINDDS trials in aggregate, statistical analysis revealed no statistically significant difference in mean age (n = 1,402, $P = 0.34$), sex frequency (n = 1,405, $P = 0.47$), or race and ethnicity frequency (n = 1,272, $P = 0.27$) between the "Before COVID-19" and "During COVID-19" groups (Fig 3). Regarding residential SES data, there was no statistically significant difference in the median ZCTA MAHI (n = 1,397, $P = 0.05$), median ZCTA MEAL (n = 1,398, $P = 0.51$), or mean ZCTA unemployment rate (n = 1,398, $P = 0.64$) between the "Before COVID-19" and "During COVID-19" groups (Fig 4).

### Patient survey data

This study analyzed responses to questions from the healthcare experience and trial satisfaction sections of the NO/AKI trial patient survey. In total, we approached 186 patients and collected 92 surveys, including 74 from patients in the "Before COVID-19" group and 18 from

**Table 1. Demographic characteristics of approached eligible patients for the NO/AKI and MINDDS trials.**

| Demographic Characteristic | Consented | Refused | Total |
|---|---|---|---|
| *Age (total n = 1,402)* | | | |
| Mean (SD), years | 68.41 (7.85) | 69.34 (7.81) | 68.81 (7.84) |
| *Sex (total n = 1,405)* | | | |
| Male (%) | 583 (74) | 388 (63) | 971 (69) |
| Female (%) | 210 (26) | 224 (37) | 434 (31) |
| *Race and ethnicity (total n = 1,272)[a]* | | | |
| Non-Hispanic white (%) | 707 (97) | 504 (93) | 1,211 (95) |
| Non-Hispanic Black (%) | 4 (1) | 10 (2) | 14 (1) |
| Hispanic (%) | 4 (1) | 11 (2) | 15 (1) |
| Asian (%) | 14 (2) | 18 (3) | 32 (3) |
| *ZCTA median annual household income (MAHI) (total n = 1,397)[a]* | | | |
| <$25,000 (%) | 0 (0) | 0 (0) | 0 (0) |
| $25,000 to <$50,000 (%) | 31 (4) | 39 (6) | 70 (5) |
| $50,000 to <$75,000 (%) | 188 (24) | 180 (30) | 368 (26) |
| $75,000 to <$100,000 (%) | 243 (31) | 174 (29) | 417 (30) |
| $100,000 to <$200,000 (%) | 312 (40) | 214 (35) | 526 (38) |
| ≥$200,000 (%) | 13 (2) | 3 (1) | 16 (1) |
| Mean (SD), USD | 98,840 (36,136) | 90,943 (32,085) | 95,392 (34,636) |
| *ZCTA median educational attainment level (MEAL) (total n = 1,398)[a]* | | | |
| Less than high school (%) | 49 (6) | 57 (9) | 106 (8) |
| High school or equivalent (%) | 229 (29) | 227 (37) | 456 (33) |
| Some college (%) | 145 (18) | 75 (12) | 220 (16) |
| Two-year degree (%) | 334 (42) | 236 (39) | 570 (41) |
| Four-year degree (%) | 31 (4) | 15 (2) | 46 (3) |
| Graduate school (%) | 0 (0) | 0 (0) | 0 (0) |
| Median | Some college | Some college | Some college |
| *ZCTA unemployment rate (total n = 1,398)* | | | |
| Mean (SD), percentage | 4.03 (1.59) | 4.21 (1.80) | 4.11 (1.68) |

[a] Percentages may not sum to 100% due to rounding

patients in the "During COVID-19" group for response rates of 49% and 50%, respectively. The average number of questions answered was 13, with a range of six to 14.

In the healthcare experience section, most patients reported that the NO/AKI trial made them more knowledgeable about their procedure. Consistent with this finding, the free response section of question eight showed that many patients learned about the possible adverse effects that cardiac surgery and cardiopulmonary bypass can have on kidney health as well as about the potential benefits of NO as a treatment. Most patients also reported that the NO/AKI trial made them a more active part of the medial care they received. Consistent with this finding, in the free response section of question nine, some patients reported asking more questions about and being more attentive to factors that affect their health. Some patients also reported providing more feedback, feeling more in tune with their care, and having more control over their treatment. Finally, most patients also reported that participating in the NO/AKI trial reduced or did not affect their anxiety level, with only three reporting increased anxiety. A summary of these responses is provided in Table 2. Importantly, responses to the healthcare experience survey questions did not differ significantly between the "Before COVID-19" and "During COVID-19" groups (Fig 5).

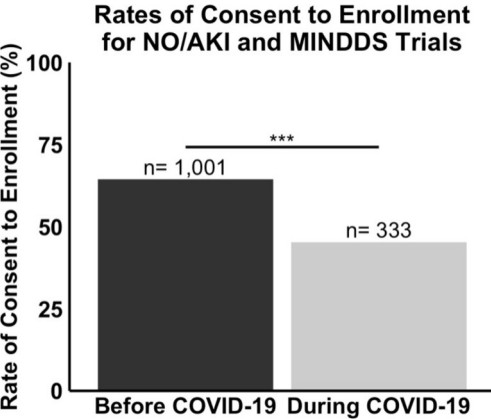

**Fig 2. The rate of consent to enrollment into the NO/AKI and MINDDS trials dropped significantly during COVID-19.** The rate of consent to enrollment among approached eligible patients was 64% (n = 1,001) in the "Before COVID-19" group and 45% (n = 333) in the "During COVID-19" group. Statistical analysis revealed a statistically significant difference in enrollment inquiry responses between the two time frames (n = 1,334, P<0.001) and a difference in proportions of 19% (95% CI 12.88% to 25.12%).

In the trial satisfaction section, most patients reported being "very satisfied" with their experience as study participants, and no patient reported dissatisfaction. Most patients also reported that they would participate in the trial again and that they would recommend the NO/AKI trial to a family member or friend if they were given the chance to participate. A summary of these responses is provided in Table 2. Importantly, responses to the trial satisfaction survey questions did not differ significantly between the "Before COVID-19" and "During COVID-19" groups (Fig 6).

## Discussion

This study had three major findings. First, the rates of consent to enrollment into the NO/AKI and MINDDS trials dropped significantly with the onset of the COVID-19 pandemic (Fig 2). Second, there were no statistically significant differences in mean age, sex frequency, race and ethnicity frequency, median ZCTA MAHI, median ZCTA MEAL, or mean ZCTA unemployment rate between the "Before COVID-19" and "During COVID-19" groups (Figs 3 and 4). And third, most respondents of the NO/AKI trial patient survey reported being satisfied with their participation and that their participation had positively impacted their healthcare experience (Table 2), and these findings did not change significantly with the onset of the COVID-19 pandemic (Figs 5 and 6).

Regarding the first finding, studies have shown that increased age, female sex, Black race, lower income, and lower education level are all associated with a decreased willingness to enroll in clinical trials [19–21]. Thus, it is plausible that the drop in rates of consent to enrollment seen in the current study could be explained by differences in these measures between the two time periods. Alternatively, this drop could also be explained by poor patient experience during the pandemic. However, given the second and third findings mentioned above, neither of these are likely the case. It is instead more likely that there were other factors contributing to the drop in rates of consent to enrollment that were simply not directly measured in the current study.

One possible explanation for the observed drop is that patients who came to the hospital may have tried to minimize contact with people as much as possible due to perceived or actual risk of contracting COVID-19. In line with this idea, studies have shown that patients actively avoided routine medical care during the COVID-19 pandemic, even if they had a known

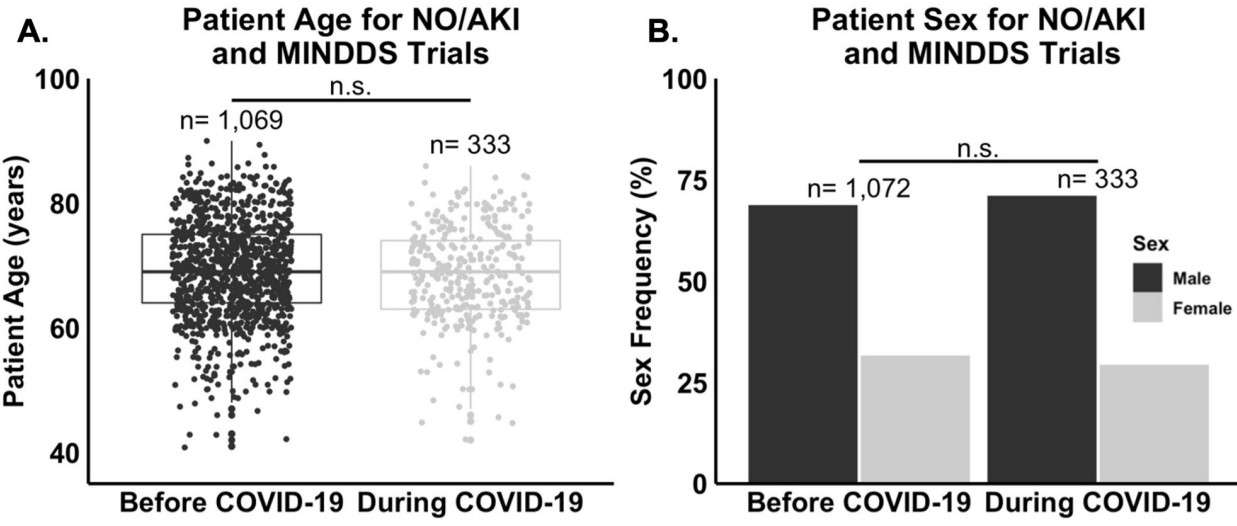

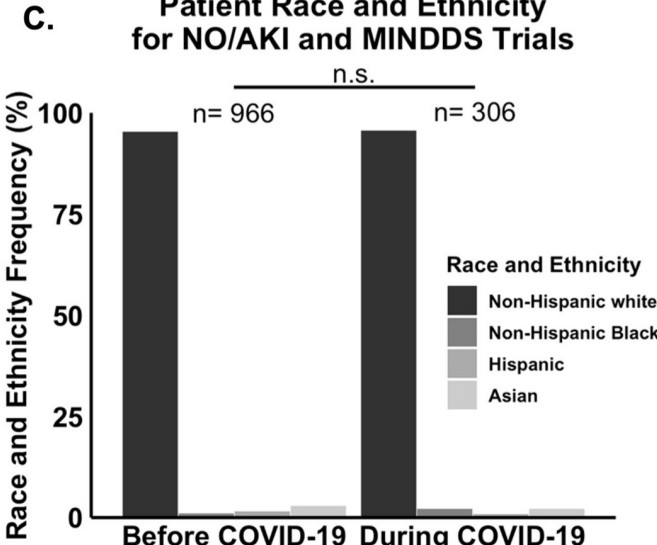

**Fig 3. General demographic factors did not differ significantly between the "Before COVID-19" and "During COVID-19" groups of the NO/AKI and MINDDS trials. (A.)** There was no statistically significant difference in mean age (n = 1,402, P = 0.34), **(B.)** sex frequency (n = 1,405, P = 0.47), or **(C.)** race and ethnicity frequency (n = 1,272, P = 0.27) between the two time frames.

underlying disease [22, 23]. In fact, some studies have shown that patients even delayed or avoided seeking emergency medical care during this time period [24–26]. A second possible explanation is that patients may have become less trusting of science during the COVID-19 pandemic and extended that mistrust into clinical trial participation. In line with this idea, research has demonstrated a general mistrust of the scientific process regarding the development of the COVID-19 vaccines [27]. Thus, it is plausible that the drop in rates of consent to enrollment seen in the current study could at least partially be explained by increased avoidance of medical care and/or increased mistrust of science during the COVID-19 pandemic.

Regarding factors that are not directly related to the patients themselves, one possible explanation for the observed drop in rates of consent to enrollment is that the attitude of research staff approaching eligible patients could have changed during the pandemic. It is well known

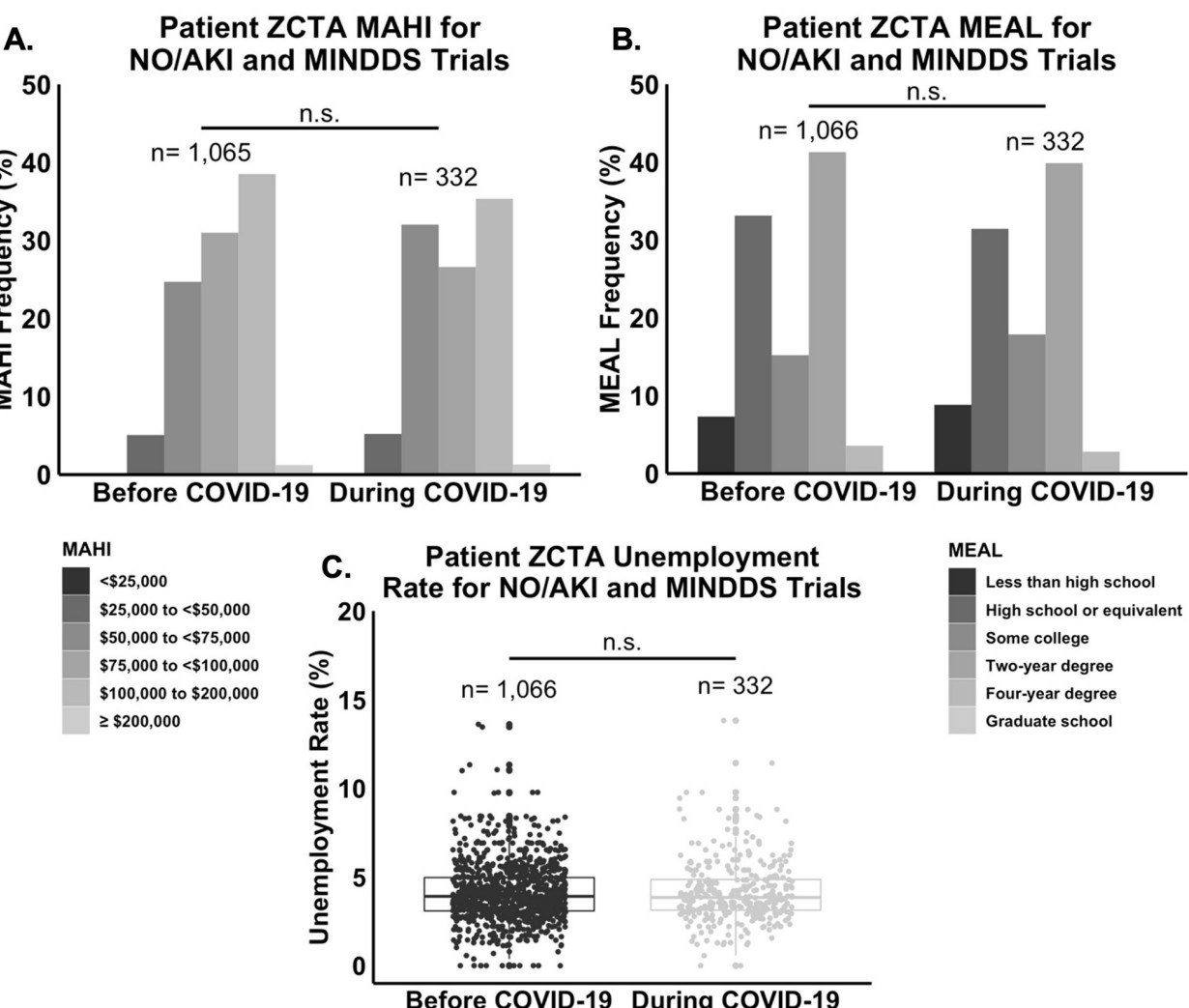

**Fig 4. Residential SES indicators did not differ significantly between the "Before COVID-19" and "During COVID-19" groups of the NO/AKI and MINDDS trials. (A.)** There was no statistically significant difference in the median ZCTA MAHI (n = 1,397, P = 0.05), **(B.)** median ZCTA MEAL (n = 1,398, P = 0.51), or **(C.)** mean ZCTA unemployment rate (n = 1,398, P = 0.64) between the two time frames.

that healthcare workers have high rates of burnout, and studies have shown that the COVID-19 pandemic heightened existing challenges that physicians face such as increasing workload, which is directly related to increased burnout [28]. Many of the research staff involved in patient recruitment for the NO/AKI and MINDDS trials both before and during the pandemic were physicians with active clinical practices, so it is possible that they experienced increased burnout and that this influenced their interactions with patients in such a way that made patients more likely to refuse clinical trial enrollment. Finally, the COVID-19 pandemic brought with it the mass usage of face masks by healthcare workers. Although studies have shown that the use of face masks does not affect the patient-doctor relationship [29, 30], some studies have shown that face masks can have a negative effect on patient-doctor communication, especially in those who are hearing impaired [31]. Thus, the possible explanations for the observed drop in rates of consent to enrollment are multifactorial, and further research should be done to parse them apart.

**Table 2. NO/AKI trial patient survey responses.**

| Survey Question | Response Frequency |
|---|---|
| *Healthcare Experience* | |
| **(8)** Did this study make you more knowledgeable about the procedure you received? (n = 90) | |
| Yes (%) | 64 (71) |
| No (%) | 26 (29) |
| **(9)** Did participating in this study make you a more active part of the medical care you received? (n = 85) | |
| Yes (%) | 51 (60) |
| No (%) | 34 (40) |
| **(10)** How did participating in the study affect your level of stress or anxiety surrounding your procedure? (n = 88) | |
| Made me a lot more anxious (%) | 0 (0) |
| Made me slightly more anxious (%) | 3 (3) |
| Did not change my anxiety level (%) | 58 (66) |
| Made me slightly less anxious (%) | 14 (16) |
| Made me a lot less anxious (%) | 13 (15) |
| *Trial Satisfaction* | |
| **(11)** How satisfied or dissatisfied were you with your experience as a study participant? (n = 91) | |
| Very dissatisfied (%) | 0 (0) |
| Dissatisfied (%) | 0 (0) |
| Neutral (%) | 0 (0) |
| Satisfied (%) | 23 (25) |
| Very satisfied (%) | 68 (75) |
| **(12)** Knowing what you know now, would you participate in this study again? (n = 91) | |
| Yes (%) | 90 (99) |
| No (%) | 1 (1) |
| **(13)** Would you recommend this study to a family member or a friend if they were given the chance to participate? (n = 91) | |
| Yes (%) | 88 (97) |
| No (%) | 3 (3) |

If these explanations are true, work should be done to address them going forward. Regarding health safety concerns, hospital staff should continue using personal protective equipment when interacting with patients, and patients should be made aware of the precautions hospitals have taken to ensure their safety. Regarding increased mistrust of the scientific process, more work should be done to make science and scientific advancements more digestible to the lay public. Patient outreach groups could aim to launch science education campaigns, understandable to anyone regardless of educational attainment. Additionally, more groups could work toward increasing the transparency of scientific development, reassuring people that uncertainty is an expected and even welcome feature of the scientific method. Addressing the issue of physician burnout is a more complicated subject, and changes aimed at decreasing work-related stresses and promoting healthy coping mechanisms would need to occur at an institutional level to have meaningful effect. Finally, healthcare workers should receive training to ensure that the use of face masks does not significantly impede the quality of their communication with patients, especially with those who are hearing impaired.

Regarding the second finding, it is not surprising that there were no statistically significant differences in any of the patient demographic or SES factors that were measured between the

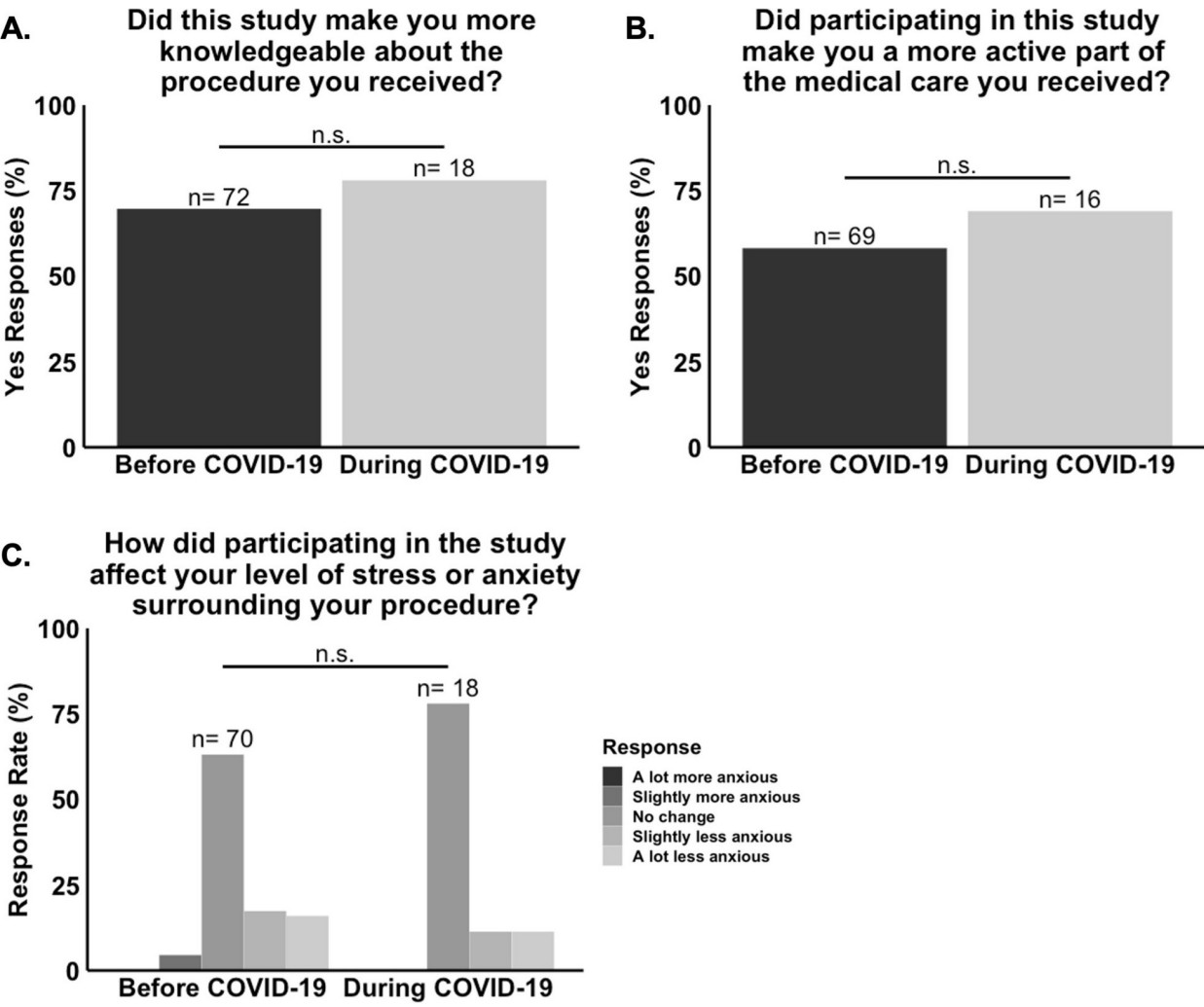

**Fig 5. Responses to the healthcare experience survey questions did not differ significantly between the "During COVID-19" and "Before and COVID-19" groups of the NO/AKI trial. (A.)** There was no statistically significant association between responses to question eight (n = 90, P = 0.68) or **(B.)** question nine (n = 85, P = 0.61) and time frame. **(C.)** There was no statistically significant difference in the median response to question 10 (n = 88, P = 0.58) between the two time frames.

"Before COVID-19" and "During COVID-19" groups (Figs 3 and 4). It has been well documented that, on average, traditionally disadvantaged patients, such as those who are from poorer backgrounds or racial and ethnic minority groups, have more challenges with accessing quality healthcare, especially during crises such as the COVID-19 pandemic. However, the patient population of the current study was largely composed of older, college educated, wealthy, non-Hispanic white males (Table 1). Therefore, it is possible, and even likely, that our negative findings regarding differences in demographic or SES factors are due to there being so few traditionally disadvantaged patients in our study population to begin with. In other words, if our study population had included more traditionally disadvantaged patients in the "Before COVID-19" group, it is possible that we would have observed more statistically significant differences in demographic or SES factors between the two time frames since these patients would have theoretically had more challenges with accessing healthcare and therefore with showing up to the hospital for inclusion in the NO/AKI and MINDDS trials after the start of the COVID-19 pandemic.

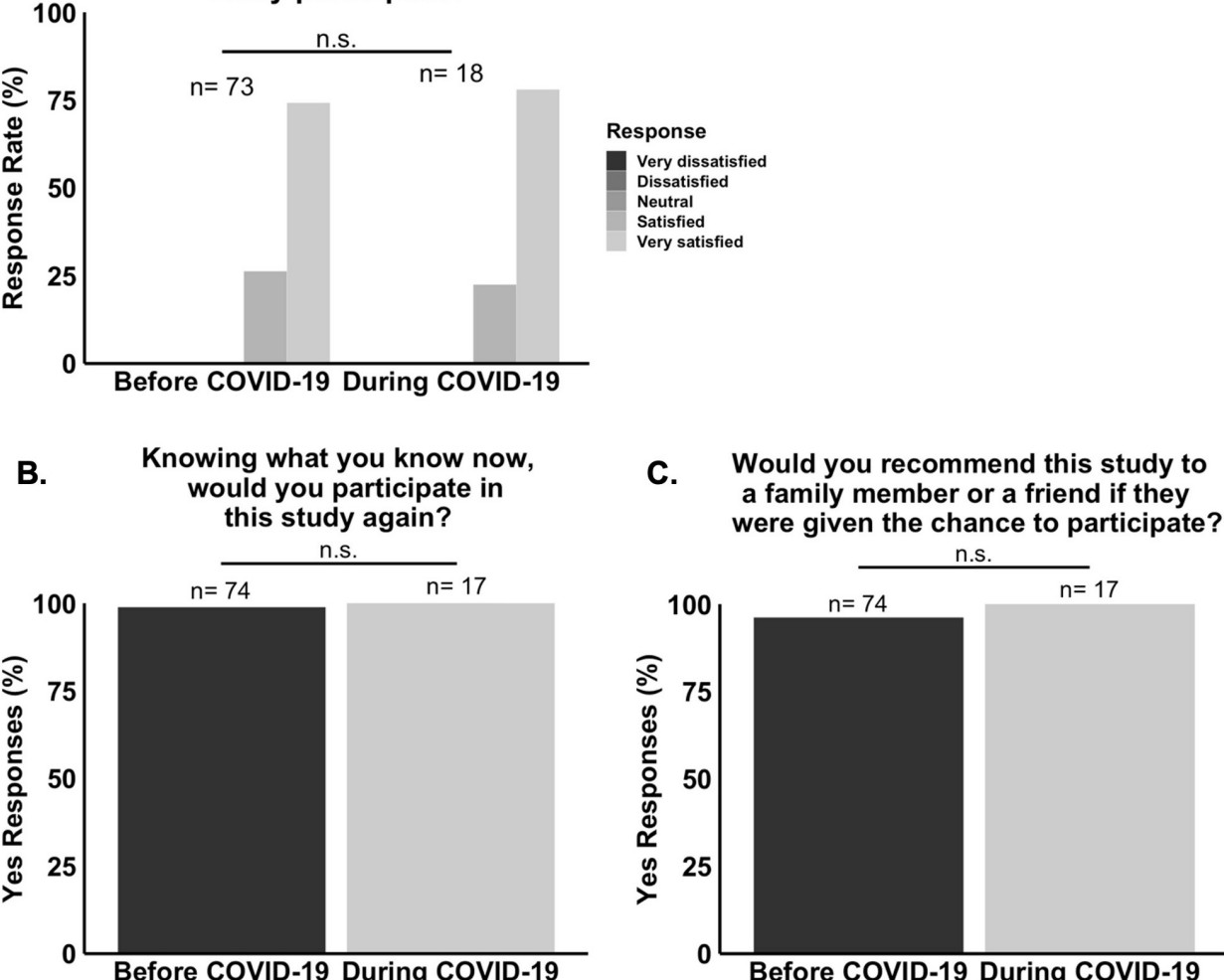

**Fig 6. Responses to the trial satisfaction survey questions did not differ significantly between the "During COVID-19" and "Before and COVID-19" groups of the NO/AKI trial. (A.)** There was no statistically significant difference in the median response to question 11 (n = 91, P = 0.75) between the two time frames. **(B.)** There was no statistically significant association between responses to question 12 (n = 91, P = 1) or **(C.)** question 13 (n = 91, P = 1) and time frame.

Finally, regarding the third finding, although the onset of the COVID-19 pandemic brought with it a drop in rates of consent to enrollment into the NO/AKI and MINDDS trials, patients who did choose to participate during the pandemic still reported being just as satisfied with their participation and continued to report that their participation had positively impacted their healthcare experiences (Figs 5 and 6). This is important because it shows that it is unlikely that the observed drop in rates of consent to enrollment was due to actual poorer patient experiences during the pandemic. However, it is still possible that more patients during the pandemic assumed or perceived that the quality of their experience would be worse than those from before the pandemic since the survey did not assess reasons for non-participation among those who refused clinical trial enrollment.

The current study was not without its limitations. First, since patients from both the NO/AKI and MINDDS trials were not asked to share information on SES indicators upon screening for clinical trial enrollment, area-level SES indicators were gathered from census data rather than from individual patients. Although studies have shown that area-level SES indicators are strongly associated with patient-level SES indicators [32], using surrogates for measures of individual SES could have limited the interpretation of our results [33, 34]. Second, inferences pertaining to traditionally disadvantaged groups were limited by the small proportion of these individuals included in the NO/AKI and MINDDS trials. Thus, future research should explore the effects of the pandemic on clinical trial participation in populations with higher proportions of patients from traditionally disadvantaged groups. Finally, although there are several plausible explanations for the observed decrease in rates of consent to enrollment into the NO/AKI and MINDDS trials with the onset of the COVID-19 pandemic, patient motives for refusal were not directly assessed in the current study and neither were attitudes of research staff approaching eligible patients or quality of communication with patients while wearing surgical masks. Further research in these areas could help definitively establish explanations for this phenomenon and thereby better inform public and institutional level efforts aimed at increasing clinical trial participation rates in the future.

## Conclusion

The COVID-19 pandemic has caused tremendous disruptions to non-COVID-19 clinical research. This study found that rates of consent to enrollment into the NO/AKI and MINDDS trials dropped significantly with the onset of the COVID-19 pandemic. Given that there were no statistically significant differences in patient demographic or SES factors between the two time periods studied, two possible explanations for this finding include increased avoidance of medical care and/or increased mistrust of science during the COVID-19 pandemic. Other possible explanations include changes in the attitudes of research staff approaching eligible patients to gain consent to enrollment and increased usage of face masks by healthcare workers. Of note, the absence of statistically significant differences in patient demographic or SES factors between the two time periods studied is unsurprising given the lack of inclusion of traditionally disadvantaged patients in our study population. Finally, this study found that patients who consented to enroll into the NO/AKI trial during the pandemic reported being just as satisfied with their participation and continued to report that their participation had positively impacted their healthcare experiences, highlighting that the observed drop in rates of consent to enrollment is unlikely to be due to actual poorer patient experiences during the pandemic. In any case, clinical trial disruptions have detrimental effects on the advancement of clinical science, patient health, and patient healthcare experience, and increased patient hesitancy to enroll in clinical trials only exacerbates these effects. Thus, understanding and addressing the issue of decreased clinical trial participation during the COVID-19 pandemic is crucial, especially as the COVID-19 pandemic continues to ravage parts of the world and we prepare for the possibility of future pandemics.

## Supporting information

**S1 Table. NO/AKI trial patient survey questions and response types.**
(DOCX)

**S1 Data. MINDDS trial demographics data.**
(XLSX)

**S2 Data. MINDDS trial enrollment data.**
(XLSX)

**S3 Data. NOAKI trial demographics data.**
(XLSX)

**S4 Data. NOAKI trial enrollment data.**
(XLSX)

**S5 Data. NOAKI trial survey data.**
(XLSX)

**S1 File. Demographics statistical analysis.**
(R)

**S2 File. Enrollment statistical analysis.**
(R)

**S3 File. NOAKI trial survey statistical analysis.**
(R)

## Author Contributions

**Conceptualization:** Josue D. Chirinos, Joseph F. Swingle, Oluwaseun Akeju, Lorenzo Berra.

**Data curation:** Josue D. Chirinos, Isabella S. Turco, Raffaele Di Fenza, Stefano Gianni, Grant M. Larson, Oluwaseun Akeju, Lorenzo Berra.

**Formal analysis:** Josue D. Chirinos, Joseph F. Swingle.

**Funding acquisition:** Lorenzo Berra.

**Investigation:** Josue D. Chirinos, Isabella S. Turco, Grant M. Larson, Oluwaseun Akeju, Lorenzo Berra.

**Methodology:** Josue D. Chirinos, Isabella S. Turco, Raffaele Di Fenza, Stefano Gianni, Grant M. Larson, Joseph F. Swingle, Oluwaseun Akeju, Lorenzo Berra.

**Project administration:** Josue D. Chirinos, Oluwaseun Akeju, Lorenzo Berra.

**Resources:** Josue D. Chirinos, Grant M. Larson, Oluwaseun Akeju, Lorenzo Berra.

**Software:** Josue D. Chirinos, Isabella S. Turco.

**Supervision:** Josue D. Chirinos, Oluwaseun Akeju, Lorenzo Berra.

**Validation:** Josue D. Chirinos, Isabella S. Turco, Raffaele Di Fenza, Stefano Gianni, Grant M. Larson, Joseph F. Swingle, Oluwaseun Akeju, Lorenzo Berra.

**Visualization:** Josue D. Chirinos, Raffaele Di Fenza, Stefano Gianni, Grant M. Larson, Oluwaseun Akeju, Lorenzo Berra.

**Writing – original draft:** Josue D. Chirinos, Isabella S. Turco.

**Writing – review & editing:** Josue D. Chirinos, Isabella S. Turco, Raffaele Di Fenza, Stefano Gianni, Grant M. Larson, Joseph F. Swingle, Oluwaseun Akeju, Lorenzo Berra.

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
