## [Decision Letter · Decision Letter 0]

28 Oct 2022

PONE-D-22-17604

Patient hesitancy in perioperative clinical trial enrollment during the COVID-19 pandemic

PLOS ONE

Dear Dr. Berra,

Thank you for submitting your manuscript to PLOS ONE. After careful consideration, we feel that it has merit but does not fully meet PLOS ONE’s publication criteria as it currently stands. Therefore, we invite you to submit a revised version of the manuscript that addresses the points raised during the review process.

ACADEMIC EDITOR:

Dear Authors,

Please carefully review the reviewers' comments and provide a point-by-point response to reviewers and submit a revised version of the manuscript.

Regarding the manuscript, a reviewer commented on the assumptions in the definition of this outcome as used in the study. Please clarify. Moreover, please add the columns for patients that refused and those that consented to enrollment. Regarding the discussion, the reviewer commented that the discussion only considered patient-related factors for refusing participation, that there was no information on the attitude of research staff approaching eligible patients, that the main limitation of the study was that there was no direct measure of the reasons for refusal, and that the discussion and conclusions were mainly speculative. Please clarify. Another reviewer suggested that a part be included in the study to show the rate of consent to enrollment in a time trend and answer whether the trend is increasing. Moreover, the importance of the study needs to be clarified. Another reviewer casted doubt on the initial anticipation of the authors when stating that patient recruitment to non-COVID-19 trials would be disrupted. Delving into more details, the reviewer states that the difference in this study is that they have studied the patient side factors. But among these factors, none of them seem to have affected their decision. I think, their positive and negative attitudes regarding participating in these studies could have been asked. Hence, those patient-side barriers could be identified and addressed in future clinical trials. 

Regards

We look forward to receiving your revised manuscript.

Kind regards,

Mohsen Abbasi-Kangevari

Academic Editor

PLOS ONE

“I have read the journal's policy and the authors of this manuscript have the following competing interests: LB has filed a patent application on June 7, 2021 for NO delivery in COVID-19 disease: PCT application number: PCT/US2021/036269.”

Additional Editor Comments:

A reviewer commented on the assumptions in the definition of this outcome as used in the study. Please clarify. Moreover, please add the columns for patients that refused and those that consented to enrollment. Regarding the discussion, the reviewer commented that the discussion only considered patient-related factors for refusing participation, that there was no information on the attitude of research staff approaching eligible patients, that the main limitation of the study was that there was no direct measure of the reasons for refusal, and that the discussion and conclusions were mainly speculative. Please clarify.

Another reviewer suggested that a part be included in the study to show the rate of consent to enrollment in a time trend and answer whether the trend is increasing. Moreover, the importance of the study needs to be clarified. Another reviewer casted doubt on the initial anticipation of the authors when stating that patient recruitment to non-COVID-19 trials would be disrupted. Delving into more details, the reviewer states that the difference in this study is that they have studied the patient side factors. But among these factors, none of them seem to have affected their decision. I think, their positive and negative attitudes regarding participating in these studies could have been asked. Hence, those patient-side barriers could be identified and addressed in future clinical trials.

Reviewers' comments:

Reviewer's Responses to Questions

**Comments to the Author**

1. Is the manuscript technically sound, and do the data support the conclusions?

Reviewer #1: Yes

Reviewer #2: Partly

Reviewer #3: Yes

Reviewer #4: Yes

Reviewer #5: Yes

2. Has the statistical analysis been performed appropriately and rigorously? 

Reviewer #1: Yes

Reviewer #2: No

Reviewer #3: Yes

Reviewer #4: Yes

Reviewer #5: Yes

3. Have the authors made all data underlying the findings in their manuscript fully available?

Reviewer #1: Yes

Reviewer #2: No

Reviewer #3: Yes

Reviewer #4: Yes

Reviewer #5: No

4. Is the manuscript presented in an intelligible fashion and written in standard English?

Reviewer #1: Yes

Reviewer #2: Yes

Reviewer #3: Yes

Reviewer #4: Yes

Reviewer #5: Yes

5. Review Comments to the Author

Reviewer #1: The authors provide a thorough account of their cross-sectional study investigating whether the covid-19 pandemic has had any impact on patient hesitance in perioperative clinical trail enrolment. It is very well-written and follows a logical flow. The methodology appears appropriate and robust. Results are presented concisely and the discussion discusses the findings in light of relevant literature. The use of clear tables and interesting figures is effective in conveying the results. Limitations are discussed, and although the authors do note it, I find it a shame that patient motives for refusal were not directly assessed in the current study. Appropriate conclusions are drawn, and overall, this is a useful piece of work which should set the scene for future research.

Reviewer #2: The authors report an observational study that compared the acceptance rate to participate in two clinical trials between pre- and during the COVID-19 pandemic.

Methods

-The comparison between periods is a crude comparison, did the authors correct for baseline patients characteristics?

-The hesitancy seems to be defined by the rate of refusion to participate, it would be informative to present the assumptions in the definition of this outcome as used in the study.

Results

-Table 1. Please add the columns for patients that refused and those that consented to enrollment.

-The subsection “Patient demographic data” presents all the denominators of Table 1. This information is confusing, perhaps is better to include this information in the table only.

-On page 9, lines 196-198: The authors state: “The risk ratio w 196 as 0.70 (95% CI 0.62 to 0.80, P<0.001), indicating that patients in the “Before COVID-19” group were 1.43 times as likely to consent to enrollment compared to patients in the “During COVID-19” group.”. It would be more appropriate to present the RR and the 95%CIs considering the “intervention” to the pre-COVID-19 period.

-The patient survey data should report the nonresponse rate for both periods.

Discussion

-The discussion only considers patient-related factors for refusing participation.

-There is no information on the attitude of research staff approaching eligible patients.

-As the authors correctly point out, the main limitation of the study is that there is no direct measure of the reasons for refusal.

Reviewer #3: Overall assessment:

Chirinos, et al., have assessed the effect of the COVID-19 pandemic on rates of eligible patient consent to enrollment in clinical trials. They showed that the enrolment rate dropped after the pandemic started. They also explored the factors that may contribute to decreasing the rate of consent to enrolment in clinical trials; however, they did not find significant results in this regard. On the plus side, the manuscript is written plain, organized, and clear which makes it easy and interesting to follow. As a weakness, the study did not survey the patients who did not consent to participate to elucidate the reasons behind it, and the analyses performed on the patient's demographic characteristics did not recommend any obvious reason. However, possible reasons are discussed concisely. Considering the importance of the topic, I think the manuscript could be accepted after applying revisions.

Overall, I think the manuscript needs revision.

Recommendation to authors:

I recommend authors add a part in their study to show the rate of consent to enrollment in a time trend and answer whether the trend is increasing. This could highlight whether the problem is being solved as the fear pandemic subsides over time. The trend could also show the effect of COVID waves on the rate of consent to enrollment.

Comment:

The abstract and the last paragraph of the introduction only mention the most important aim and finding of the study. I think this could cause readers to underestimate the importance of the study. I recommend changing these parts to reflect some other most important findings of the study and the studies performed to attain the findings.

Reviewer #4: The authors have successfully intended to investigate the effect of the COVID-19 pandemic on rates of eligible patient consent 87 to enrollment into two non-COVID-19 clinical trials. I do see eye to eye with the authors about the negative impact of disruption in these investigations and thus, I find this study valuable. However, there are minor comments that need to be addressed:

First of all, it is anticipated that patient recruitment to non-COVID-19 trials would be disrupted and there have been previous studies supporting this idea. The difference in this study is that they have studied the patient side factors. But among these factors, none of them seem to have affected their decision. I think, their positive and negative attitudes regarding participating in these studies could have been asked. Hence, those patient-side barriers could be identified and addressed in future clinical trials.

Please add clinical trial IDs in the methods.

What is the logic behind studying healthcare experience and trial satisfaction sections of the NO/AKI trial patient survey? It seems to fall apart from the study’s objective. Unless the authors were aiming to investigate whether COVID-19 has changed the enrolled patients’ minds regarding the trial. In that case, they can report that although patients mostly refused to participate, those who participated were still satisfied. Please make your objective and interpretation of this part clear in the introduction and the discussion. Accordingly, the third finding of this study has not been discussed at all.

Line 264: (S1 and S2 Figs S1), what is the S1 at the end?

It would be better if the authors could also try to explain why their studied variables did not have significant effects.

Reviewer #5: Dear Authors,

Thank you for giving me the opportunity to review your article about the effect of COVID-19 on patients participation in clinical trials. I found the topic interesting. The methodology of study and the process of collecting the data is appropriately explained. however, I have following feedbacks:

1. I suggest to address the exact number of patient consented to enrollment in NO/AKI and MINDDS trials separately (line 170 and Fig1).

2 Please explain why you included patients of two different trials in one sample for analysis. The analysis of two trials separately might yields different results.

3. Please report all results with confidence intervals. Especially, the rate of consent to enrollment in line 195 and 195 and Figure 2.

6. PLOS authors have the option to publish the peer review history of their article (what does this mean?). If published, this will include your full peer review and any attached files.

Reviewer #1: No

Reviewer #2: No

Reviewer #3: No

Reviewer #4: **Yes: **Mohammadreza Azangou-Khyavy

Reviewer #5: No

---

## [Author Response · Author response to Decision Letter 0]

1 Dec 2022

December 1, 2022

Re: Resubmission of manuscript entitled “Patient hesitancy in perioperative clinical trial enrollment during the COVID-19 pandemic,” (PONE-D-22-17604)

The Editors

PLOS ONE

1265 Battery Street, Suite 200

San Francisco, CA 94111

Dear Editors,

 Thank you for giving us the opportunity to revise our manuscript entitled “Patient hesitancy in perioperative clinical trial enrollment during the COVID-19 pandemic.” We appreciate the careful review and constructive suggestions. It is our belief that the manuscript is substantially improved after making the suggested edits.

 Included below in this letter are the Academic Editor and Reviewer comments with our responses in red font, including how and where the text was modified. Changes made in the manuscript were marked using track changes, and both this file and a separate identical file without track changes have been submitted along with this letter. Please note that the line numbers in the file with track changes are slightly different from those in the file without track changes due to formatting. Thus, all manuscript locations given in the outline below correspond to the line number in the file without track changes for consistency. The revision has been developed in consultation with all coauthors, and each author has given approval to the final form of this revision. The agreement form signed by each author remains valid.

Below, you will also find our updated Financial Disclosure and Competing Interests statements. Thank you for your time and consideration. We look forward to hearing back.

Financial Disclosure: LB receives salary support from K23 HL128882/NHLBI NIH as principal investigator for his work on hemolysis and nitric oxide. LB receives technologies and devices from Inhaled Nitric Oxide (NO) Therapeutics LLC, Masimo Corp. This study was supported through LB by the Reginald Jenney Endowment Chair at Harvard Medical School, by Sundry Funds at Massachusetts General Hospital, and by laboratory funds of the Anesthesia Center for Critical Care Research of the Department of Anesthesia, Critical Care, and Pain Medicine at Massachusetts General Hospital. The funders had no role in study design, data collection and analysis, decision to publish, or preparation of the manuscript.

Competing Interests statement: I have read the journal's policy and the authors of this manuscript have the following competing interests: LB has filed a patent application on June 7, 2021 for NO delivery in COVID-19 disease: PCT application number: PCT/US2021/036269. This does not alter our adherence to PLOS ONE policies on sharing data and materials. All other authors have declared that no competing interests exist.

Sincerely,

Lorenzo Berra, MD (corresponding author)

Department of Anesthesia, Critical Care, and Pain Medicine, Massachusetts General Hospital

Harvard Medical School

lberra@mgh.harvard.edu 

Journal Requirements

o Author response: We deleted author titles (title page, lines 7-8), added the country for one of the author’s affiliations (title page, line 12), and updated the formatting of the title page (title page, lines 12, 14, 17, 19, 22, 26-27) to meet PLOS ONE’s style requirements. We also changed the information on COVID-19 cases given in the first paragraph of the introduction to be as updated as possible (page 2, lines 58-59).

• We note that the grant information you provided in the “Funding Information” and “Financial Disclosure” sections do not match. When you resubmit, please ensure that you provide the correct grant numbers for the awards you received for your study in the “Funding Information” section.

o Author response: We have updated our Financial Disclosure and included it in our cover letter above. The only relevant grant number to the funding of this study is for LB’s salary support (i.e., K23 HL128882/NHLBI NIH), and we have confirmed that this grant number, included in the Financial Disclosure, matches the one listed under the Funding Information section in the submission portal (see updated Financial Disclosure in cover letter above).

• Thank you for stating the following in the Competing Interests section: “I have read the journal's policy and the authors of this manuscript have the following competing interests: LB has filed a patent application on June 7, 2021 for NO delivery in COVID-19 disease: PCT application number: PCT/US2021/036269.” Please confirm that this does not alter your adherence to all PLOS ONE policies on sharing data and materials, by including the following statement: "This does not alter our adherence to PLOS ONE policies on sharing data and materials.” (as detailed online in our guide for authors http://journals.plos.org/plosone/s/competing-interests). Please include your updated Competing Interests statement in your cover letter; we will change the online submission form on your behalf.

o Author response: We updated our Competing Interests statement to confirm that our interests do not alter our adherence to PLOS ONE policies on sharing data and materials (see updated Competing Interests statement in cover letter above).

Academic Editor (Mohsen Abbasi-Kangevari)

• A reviewer commented on the assumptions in the definition of this outcome as used in the study. Please clarify.

o See response to Reviewer 2 below

• Moreover, please add the columns for patients that refused and those that consented to enrollment.

o See response to Reviewer 2 below

• Regarding the discussion, the reviewer commented that the discussion only considered patient-related factors for refusing participation, that there was no information on the attitude of research staff approaching eligible patients, that the main limitation of the study was that there was no direct measure of the reasons for refusal, and that the discussion and conclusions were mainly speculative. Please clarify.

o See response to Reviewer 2 below

• Another reviewer suggested that a part be included in the study to show the rate of consent to enrollment in a time trend and answer whether the trend is increasing.

o See response to Reviewer 3 below

• Moreover, the importance of the study needs to be clarified.

o See response to Reviewer 3 below

• Another reviewer casted doubt on the initial anticipation of the authors when stating that patient recruitment to non-COVID-19 trials would be disrupted. Delving into more details, the reviewer states that the difference in this study is that they have studied the patient side factors. But among these factors, none of them seem to have affected their decision. I think, their positive and negative attitudes regarding participating in these studies could have been asked. Hence, those patient-side barriers could be identified and addressed in future clinical trials.

o See response to Reviewer 4 below

Reviewer 1

• The authors provide a thorough account of their cross-sectional study investigating whether the covid-19 pandemic has had any impact on patient hesitance in perioperative clinical trail enrolment. It is very well-written and follows a logical flow. The methodology appears appropriate and robust. Results are presented concisely, and the discussion discusses the findings in light of relevant literature. The use of clear tables and interesting figures is effective in conveying the results. Limitations are discussed, and although the authors do note it, I find it a shame that patient motives for refusal were not directly assessed in the current study. Appropriate conclusions are drawn, and overall, this is a useful piece of work which should set the scene for future research.

o Author response: No additional changes made.

Reviewer 2

• The authors report an observational study that compared the acceptance rate to participate in two clinical trials between pre- and during the COVID-19 pandemic.

• Methods

o The comparison between periods is a crude comparison, did the authors correct for baseline patient characteristics?

Author response: We used a two-sided t-test to compare the rates of consent to enrollment between the “Before COVID-19” and “During COVID-19” groups. Although this analysis itself does not correct for baseline patient characteristics, we followed up our t-test results with individual analyses of baseline patient characteristics we felt could potentially explain our findings (e.g., patient age, sex, race and ethnicity, etc.). In these subsequent analyses, we found that there were no statistically significant differences in any of these baseline patient characteristics. No additional changes made.

o The hesitancy seems to be defined by the rate of refusion to participate, it would be informative to present the assumptions in the definition of this outcome as used in the study.

Author response: We added a sentence to clarify the definition of the term “hesitancy” as used in the manuscript (page 3, lines 99-101).

• Results

o Table 1. Please add the columns for patients that refused and those that consented to enrollment.

Author response: We added the appropriate columns to Table 1 (pages 7-9, Table 1).

o The subsection “Patient demographic data” presents all the denominators of Table 1. This information is confusing, perhaps is better to include this information in the table only.

Author response: We deleted the information in parentheses from this paragraph and now only show denominator information in Table 1 (page 7, lines 184-187).

o On page 9, lines 196-198: The authors state: “The risk ratio was 0.70 (95% CI 0.62 to 0.80, P<0.001), indicating that patients in the “Before COVID-19” group were 1.43 times as likely to consent to enrollment compared to patients in the “During COVID-19” group.”. It would be more appropriate to present the RR and the 95%CIs considering the “intervention” to the pre-COVID-19 period.

Author response: We agree that the way in which we initially presented this data is confusing, as the “intervention group” in the risk ratio is different from the “intervention group” in the interpretation of the risk ratio. However, we believe that keeping the “During COVID-19” group as the “intervention group” makes more sense since this group was the true “intervention group” of the study. We therefore changed the interpretation of the risk ratio to a statement of percent relative effect in which the “intervention group” is the “During COVID-19” group (page 9, lines 203-204).

o The patient survey data should report the nonresponse rate for both periods.

Author response: We added the nonresponse rates for both periods (page 10, lines 234-236).

• Discussion

o The discussion only considers patient-related factors for refusing participation.

Author response: We added a discussion on how the attitude of research staff approaching eligible patients and the fact that research staff approaching eligible patients began wearing face masks during the pandemic could have contributed to increased patient refusal (page 15, lines 309-324). We also added a discussion on how these issues could be addressed in the future (page 15-16, lines 334-338). Finally, we updated the conclusion accordingly (page 18, lines 465-468).

o There is no information on the attitude of research staff approaching eligible patients.

Author response: See response to Reviewer comment above.

o As the authors correctly point out, the main limitation of the study is that there is no direct measure of the reasons for refusal.

Author response: We agree with this statement. Additionally, given the changes discussed above, we added two new limitations to our discussion, which are that the study (1) did not assess the attitude of research staff approaching eligible patients and (2) did not assess the quality of communication with patients while research staff were wearing surgical masks (page 17, lines 377-379).

Reviewer 3

• Chirinos, et al., have assessed the effect of the COVID-19 pandemic on rates of eligible patient consent to enrollment in clinical trials. They showed that the enrolment rate dropped after the pandemic started. They also explored the factors that may contribute to decreasing the rate of consent to enrolment in clinical trials; however, they did not find significant results in this regard. On the plus side, the manuscript is written plain, organized, and clear which makes it easy and interesting to follow. As a weakness, the study did not survey the patients who did not consent to participate to elucidate the reasons behind it, and the analyses performed on the patient's demographic characteristics did not recommend any obvious reason. However, possible reasons are discussed concisely. Considering the importance of the topic, I think the manuscript could be accepted after applying revisions.

• Overall, I think the manuscript needs revision. Recommendation to authors:

o I recommend authors add a part in their study to show the rate of consent to enrollment in a time trend and answer whether the trend is increasing. This could highlight whether the problem is being solved as the fear pandemic subsides over time. The trend could also show the effect of COVID waves on the rate of consent to enrollment.

Author response: We agree with the Reviewer that a graph of this kind would be interesting to see. Unfortunately, the information contained within our dataset is not granular enough to allow us to create such a graph. In fact, for many patients, we do not have exact dates that they were approached to gain consent to enrollment. Some of the dates are listed simply as months, some are listed as weeks, and still others are listed in time periods that span the end of one month and the beginning of the next. This makes it virtually impossible to calculate rates of consent to enrollment in a time trend where the time intervals between data points are both consistent between each other and small enough to capture possible changes as COVID waves progressed. In fact, this was the main reason we decided to present the data in the way that we did (i.e., comparing two groups that each span larger time periods, namely “Before COVID-19” and “During COVID-19”), as trying to get any more granular to show the data in a time trend proved to be unfruitful. If the Reviewer has any suggestions for how to present this data, we would be happy to try that in future revisions if necessary. No additional changes made.

o The abstract and the last paragraph of the introduction only mention the most important aim and finding of the study. I think this could cause readers to underestimate the importance of the study. I recommend changing these parts to reflect some other most important findings of the study and the studies performed to attain the findings.

Author response: The abstract (page 1, lines 44 and 48-51) and conclusion (page 18, lines 386-388 and 392-398) have been updated accordingly.

Reviewer 4 (Mohammadreza Azangou-Khyavy)

• The authors have successfully intended to investigate the effect of the COVID-19 pandemic on rates of eligible patient consent 87 to enrollment into two non-COVID-19 clinical trials. I do see eye to eye with the authors about the negative impact of disruption in these investigations and thus, I find this study valuable. However, there are minor comments that need to be addressed:

o First of all, it is anticipated that patient recruitment to non-COVID-19 trials would be disrupted and there have been previous studies supporting this idea. The difference in this study is that they have studied the patient side factors. But among these factors, none of them seem to have affected their decision. I think, their positive and negative attitudes regarding participating in these studies could have been asked. Hence, those patient-side barriers could be identified and addressed in future clinical trials.

Author response: We agree with this Reviewer comment, as it represents an accurate and concise summary of our paper. We discuss possible explanations for our findings in the discussion section. No further changes made.

o Please add clinical trial IDs in the methods.

Author response: We added clinical trial IDs to the methods section (page 3, lines 90 and 92).

o What is the logic behind studying healthcare experience and trial satisfaction sections of the NO/AKI trial patient survey? It seems to fall apart from the study’s objective. Unless the authors were aiming to investigate whether COVID-19 has changed the enrolled patients’ minds regarding the trial. In that case, they can report that although patients mostly refused to participate, those who participated were still satisfied. Please make your objective and interpretation of this part clear in the introduction and the discussion. Accordingly, the third finding of this study has not been discussed at all.

Author response: We agree that the reason for including this data was not made very clear in the original manuscript. We have therefore clarified our objective regarding inclusion of this data by adding sentences to the introduction (page 3, lines 81-83) and methods (page 5, lines 129-130 and 132-134) sections. Additionally, to highlight the importance of this data, we have changed the associated figures from supplemental figures to regular figures. Thus, S1 Fig is now Fig 5 (page 11, lines 251-258) and S2 Fig is now Fig 6 (page 13, lines 268-275). Figure legends were removed from the Supporting information section accordingly (page 22, line 502). Finally, we have added to our interpretation of this data and discussed the third finding at greater length in the discussion section (pages 16-17, lines 355-364).

o Line 264: (S1 and S2 Figs S1), what is the S1 at the end?

Author response: This was a typo, as the S1 at the end of “(S1 and S2 Figs S1)” should not have been there. However, we changed “S1 Fig” to “Fig 5” and “S2 Fig” to Fig 6 (see response to Reviewer comment above), and corrective changes were made to these figure names accordingly.

o It would be better if the authors could also try to explain why their studied variables did not have significant effects.

Author response: We added a paragraph to the discussion section where we discuss possible reasons for why our studied variables did not have significant effects (page 16, lines 339-354).

Reviewer 5

• Thank you for giving me the opportunity to review your article about the effect of COVID-19 on patient participation in clinical trials. I found the topic interesting. The methodology of study and the process of collecting the data is appropriately explained. However, I have following feedbacks:

o I suggest addressing the exact number of patients consented to enrollment in the NO/AKI and MINDDS trials separately (line 170 and Fig1).

Author response: See response to Reviewer comment below.

o Please explain why you included patients of two different trials in one sample for analysis. The analysis of two trials separately might yields different results.

Author response: The main reason we did not separate out the two clinical trials was that we felt like doing so drew too much focus on comparing them, which was not our main objective and also out of the scope of our study. Additionally, the two clinical trial samples were very similar to one another (e.g., they were drawn from the same patient population as outlined in the methods section, they were drawn from very similar and overlapping time frames, they were approached by similar research teams, etc.), and preliminary analysis showed that our findings were no different whether we separated or combined the data. Thus, for conciseness, to increase the robustness of our data, and to avoid turning our study into a comparative one, we decided to present the data in a combined fashion. We would strongly prefer to keep our data this way, but if this represents a non-negotiable suggestion, we will consider separating out the analysis in future revisions. No additional changes made.

o Please report all results with confidence intervals. Especially, the rate of consent to enrollment in line 195 and 195 and Figure 2.

Author response: We calculated a difference in proportions for the rate of consent to enrollment. We also added a confidence interval for this estimate in the results section (page 9, lines 200-201 and 211-212).

Comments to the Author

1. Is the manuscript technically sound, and do the data support the conclusions?

• Reviewer #1: Yes

• Reviewer #2: Partly

o Author response: See responses to Reviewer 2 above. We believe that our changes improve the manuscript in such a way as to meet the requirements laid out by this question.

• Reviewer #3: Yes

• Reviewer #4: Yes

• Reviewer #5: Yes

2. Has the statistical analysis been performed appropriately and rigorously?

• Reviewer #1: Yes

• Reviewer #2: No

o Author response: See responses to Reviewer 2 above. We believe that our changes improve the manuscript in such a way as to meet the requirements laid out by this question.

• Reviewer #3: Yes

• Reviewer #4: Yes

• Reviewer #5: Yes

3. Have the authors made all data underlying the findings in their manuscript fully available?

• Reviewer #1: Yes

• Reviewer #2: No

o Author response: Upon initial submission of our manuscript, we uploaded four items as Supplemental Information. These four items included all of our raw enrollment data as well as our raw data on patient demographic and SES factors. With our revisions, we additionally uploaded our raw NO/AKI trial survey data as well as all of our code for our statistical analyses in R. We believe that these changes improve the manuscript in such a way as to meet the requirements laid out by this question.

• Reviewer #3: Yes

• Reviewer #4: Yes

• Reviewer #5: No

o Author response: See response to Reviewer 2 above.

4. Is the manuscript presented in an intelligible fashion and written in standard English?

• Reviewer #1: Yes

• Reviewer #2: Yes

• Reviewer #3: Yes

• Reviewer #4: Yes

• Reviewer #5: Yes

---

## [Decision Letter · Decision Letter 1]

12 Dec 2022

Patient hesitancy in perioperative clinical trial enrollment during the COVID-19 pandemic

PONE-D-22-17604R1

Dear Dr. Berra,

We’re pleased to inform you that your manuscript has been judged scientifically suitable for publication and will be formally accepted for publication once it meets all outstanding technical requirements.

Kind regards,

Seth Kwabena Amponsah, PhD

Academic Editor

PLOS ONE

Additional Editor Comments (optional):

Reviewers' comments:

Reviewer's Responses to Questions

**Comments to the Author**

1. If the authors have adequately addressed your comments raised in a previous round of review and you feel that this manuscript is now acceptable for publication, you may indicate that here to bypass the “Comments to the Author” section, enter your conflict of interest statement in the “Confidential to Editor” section, and submit your "Accept" recommendation.

Reviewer #2: All comments have been addressed

Reviewer #3: All comments have been addressed

Reviewer #4: All comments have been addressed

2. Is the manuscript technically sound, and do the data support the conclusions?

Reviewer #2: Yes

Reviewer #3: Yes

Reviewer #4: Yes

3. Has the statistical analysis been performed appropriately and rigorously? 

Reviewer #2: Yes

Reviewer #3: Yes

Reviewer #4: Yes

4. Have the authors made all data underlying the findings in their manuscript fully available?

Reviewer #2: Yes

Reviewer #3: Yes

Reviewer #4: No

5. Is the manuscript presented in an intelligible fashion and written in standard English?

Reviewer #2: Yes

Reviewer #3: Yes

Reviewer #4: Yes

6. Review Comments to the Author

Reviewer #2: In the revised version of the manuscript, the authors have appropriately addressed all comments and made changes consequently.

Reviewer #3: Dear Madam/Sir,

Considering the limitations authors mentioned on the access to the exact dates that patients were approached to gain consent to enrollment, I think adding the time-trend I recommended would not be possible for authors. Therefore, I think the authors addressed my comments and the paper is acceptable.

Regards

Reviewer #4: (No Response)

7. PLOS authors have the option to publish the peer review history of their article (what does this mean?). If published, this will include your full peer review and any attached files.

Reviewer #2: **Yes: **Javier Mariani

Reviewer #3: No

Reviewer #4: **Yes: **Mohammadreza Azangou-Khyavy

---

## [Editor Report · Acceptance letter]

4 Jan 2023

PONE-D-22-17604R1 

Patient hesitancy in perioperative clinical trial enrollment during the COVID-19 pandemic 

Dear Dr. Berra:

I'm pleased to inform you that your manuscript has been deemed suitable for publication in PLOS ONE. Congratulations! Your manuscript is now with our production department. 

Kind regards, 

on behalf of

Dr. Seth Kwabena Amponsah 

Academic Editor

PLOS ONE